# UV-Excited Luminescence in Porous Organosilica Films with Various Organic Components

**DOI:** 10.3390/nano13081419

**Published:** 2023-04-20

**Authors:** Md Rasadujjaman, Jinming Zhang, Dmitry A. Spassky, Sergej Naumov, Alexey S. Vishnevskiy, Konstantin A. Vorotilov, Jiang Yan, Jing Zhang, Mikhail R. Baklanov

**Affiliations:** 1Department of Microelectronics, North China University of Technology, Beijing 100144, China; jinming@naura.com (J.Z.); jiangyan@ncut.edu.cn (J.Y.); baklanovmr@gmail.com (M.R.B.); 2Department of Physics, Mawlana Bhashani Science and Technology University, Santosh, Tangail 1902, Bangladesh; 3Skobeltsyn Institute of Nuclear Physics, Lomonosov Moscow State University, Moscow 119991, Russia; deris2002@mail.ru; 4Institute of Physics, University of Tartu, 50411 Tartu, Estonia; 5Leibniz Institute of Surface Engineering (IOM), 04318 Leipzig, Germany; sergej.naumov@iom-leipzig.de; 6Research and Education Center “Technological Center”, MIREA—Russian Technological University (RTU MIREA), Moscow 119454, Russia; alexeysw@mail.ru (A.S.V.); vorotilov@live.ru (K.A.V.); 7European Centre for Knowledge and Technology Transfer (EUROTEX), 1040 Brussels, Belgium

**Keywords:** low-k dielectrics, organosilica glass, interconnects, photoluminescence, oxygen deficient centers

## Abstract

UV-induced photoluminescence of organosilica films with ethylene and benzene bridging groups in their matrix and terminal methyl groups on the pore wall surface was studied to reveal optically active defects and understand their origin and nature. The careful selection of the film’s precursors and conditions of deposition and curing and analysis of chemical and structural properties led to the conclusion that luminescence sources are not associated with the presence of oxygen-deficient centers, as in the case of pure SiO_2_. It is shown that the sources of luminescence are the carbon-containing components that are part of the low-k-matrix, as well as the carbon residues formed upon removal of the template and UV-induced destruction of organosilica samples. A good correlation between the energy of the photoluminescence peaks and the chemical composition is observed. This correlation is confirmed by the results obtained by the Density Functional theory. The photoluminescence intensity increases with porosity and internal surface area. The spectra become more complicated after annealing at 400 °C, although Fourier transform infrared spectroscopy does not show these changes. The appearance of additional bands is associated with the compaction of the low-k matrix and the segregation of template residues on the surface of the pore wall.

## 1. Introduction

Porous organosilica films have many different applications, from catalysis and drug and gene delivery to microelectronics [1]. One of their most economically significant applications is with ultra-large-scale integration (ULSI) devices where porous low-dielectric constant (low-k) materials are used to reduce signal propagation delay in metallization wires [2]. However, the integration of porous low-k materials with metal wires faces numerous challenges, and the most important problems are related to the degradation of their dielectric properties and reliability. The most studied factors that degrade leakage current and reliability are related to structural modifications and adsorbed moisture in “plasma-damaged” low-k dielectrics and hydrocarbon residues formed during sacrificial porogen removal [3,4,5,6,7,8,9,10,11,12,13,14,15]. Photon irradiation increases the intrinsic defect density and creates trapped charges inside the low-k material, which can lead to reliability issues [16,17]. The knocking-off of atoms from the low-k material network can also occur during the ion sputtering process leading to the formation of Si vacancies such as EX centers or dangling carbon bonds, where the carbon-related defects contribute to higher leakage [18]. In addition, the formation of surface oxygen vacancies, probably due to the removal of terminal organic groups after Ar^+^ sputtering, leads to the formation of sub-gap surface states at 5.0 and 7.2 eV [18]. Atomic defects such as non-bridging oxygen hole centers (NBOHC) and oxygen vacancies (E’centers) have been studied by electron spin resonance (ESR) spectroscopy, and the results of these studies of low-k materials are discussed and summarized in Refs. [17,19,20]. Recently, the UV-induced photoluminescence (PL) of a mesoporous organosilica low-k dielectric was studied [21], and it was concluded that the formation of oxygen-deficient centers (ODCs), ODC(I) (≡Si–Si≡) and ODC(II) (=Si:) centers, similar to those observed in pure SiO_2_ [22,23] can explain the leakage current mechanism studied in Refs. [24,25]. However, it is not always easy to distinguish between the influence of atomic defects and residual carbon on the critical properties of organosilicate glass (OSG) low-k dielectrics.

So far, most defect studies have been performed with so-called methyl-terminated organosilica glasses: silica-like materials, where some oxygen bridging atoms in the silica matrix are replaced by two methyl groups (≡Si–O–Si≡ → ≡Si–CH_3_ … CH_3_–Si≡). This reduces the matrix density but makes the films sufficiently hydrophobic. These materials are mainly deposited by plasma-enhanced chemical vapor deposition (PECVD) [26]. However, the need to improve the mechanical properties and reliability of low-k dielectrics stimulated the extensive study of materials with different types of carbon bridges between silicon atoms [27,28,29,30,31,32,33,34]. Replacing the oxygen bridge with carbon allows for a significant improvement in mechanical properties due to the higher bending rigidity of the ≡Si–C–Si≡ bonds compared to the ≡Si–O–Si≡ bonds [31]. Evaporation-induced self-assembly (EISA) [35] using carbon-bridged alkoxysilane precursors has been shown to produce periodic mesoporous organosilica (PMO) with ordered porosity and hydrocarbon bridges in the film matrix are formed. Their properties, including the thermal and chemical resistance of different carbon bridges, have been extensively evaluated [36,37,38].

The present research aims to study the origin of ultra-violet (UV)-induced PL in porous organosilica films containing various organic components. The films deposited had a well-defined chemical composition and porosity. Using materials with different and controlled compositions and porosities (Figure 1), we hoped to understand the physical nature of the optically and electrically active defects.

Three different types of organosilica glasses were used in this research. The first type (1-1, 1-2, 2-2, 2-3) includes periodic mesoporous organosilica with ethylene and 1,4-benzene bridges in their matrix. The second type (2-1) film includes a recently proposed “hyperconnected structure” based on 1,3,5- and 1,3-benzene bridges [33]. The third type of film (3-1, 3-2) has a “classic” low-k structure with methyl terminal groups and random porosity [2,3]. The conditions of their preparation and properties are described in the experimental part and in the Appendix A, and even more detailed information can be found in the original publications cited.

## 2. Materials and Methods

### 2.1. Material Preparation

i. The first type of sample includes ethylene-bridged PMO materials (Figure 1). The precursor solutions were prepared by co-hydrolysis of alkylenealkoxysilane—1,2-bis(triethoxysilyl)methane (BTMSE) with alkylalkoxysiloxane—methyltrimethoxysilane (MTMS) (Fluka, Buchs, Switzerland) under acidic conditions as described in detail in Refs. [39,40]. Due to the presence of both BTMSE and MTMS, the film contains both ethylene bridges and methyl terminal groups (Figure 1). This approach is used for low-k film preparation since the materials containing only a carbon bridge do not have sufficient hydrophobicity. Conversely, materials with only terminal methyl groups have insufficiently good mechanical properties due to the reduction in matrix connectivity. All films were deposited on Si (100) wafers, and film preparation was completed by a two-step annealing process: (1) soft-bake at 150 °C for 30 min on a hot plate to remove the solvent, (2) hard-bake at 400 °C for 30 min in a dry air oven. After deposition and curing, the wafers were cut for analysis by different techniques.

To prepare the films with different porosity but the same matrix composition, the matrix solution was divided into four parts, and different amounts of Brij^®^30 surfactant (Sigma-Aldrich, St. Louis, MO, USA) were added and stirred. The added surfactant acts as a porogen, directing the structure forming a micelle-forming structure and interacting with the precursor. After the bulk structure was assembled, the surfactant was removed, leaving pores or channels embedded in the material framework. The formation of hybrid OSG films is based on the hydrolysis and condensation of MTMS and BTMSE precursors:≡Si–OCH_3_ + H_2_O → ≡Si–OH + CH_3_–OH (hydrolysis)(1)
≡Si–OH + HO–Si≡ → ≡Si–O–Si≡ +H_2_O (condensation)(2)

As-deposited (AD) films contain silanol groups formed as a result of precursor hydrolysis (Reaction (1)) and some unreacted methoxy (–OCH_3_) groups from precursors. The concentration of these groups can still be significant after the soft bake at 150 °C (SB) but almost not detectable after the final curing at 400 °C (HB). SB films mainly reflect the non-relaxed structure determined by reactions (1) and (2). HB increases the degree of cross-linking of the organosilicate matrix with a significant improvement in the bonding structure of the Si–O–Si network [8,41]. At this stage, the matrix compaction occurs, accompanied by the segregation of unassembled porogen residues from the low-k matrix. Photoluminescence was measured from AD samples, SB and HB samples. Samples termed “as-deposited (AD)” were dried at room temperature and 60–80 °C for 10 min. Destruction of the ethylene bridge during AD and SB is not expected, and these samples cannot contain ODC centers.

ii. Pure 1,4-benzene-bridged films (2-2 and 2-3 in Figure 1) were deposited using 1,4-bis(triethoxysilyl)benzene (BTESB). The “dense” version of the films had an EP-measured porosity (free volume) of about 10% (1,4-BB), while the sample deposited with 30 wt% porogen had a porosity of about 29% (1,4-BB-p). In contrast to the ethylene-bridged and methyl-terminated samples, the benzene-bridged films had an extremely small pore size of about 0.5 nm against 3 nm in MTMS-p (3-2) samples. It also had much better mechanical properties, as demonstrated in the paper [42]. Furthermore, all the details related to the film deposition, including curing conditions and measurement procedures, can be found in Ref. [42].

iii. Recently, hyperconnected structures developed by groups at IBM and Stanford University have attracted particular interest [33]. These films were deposited using a mixture of 1,3,5-tris(triethoxysilyl)benzene and 1,3-bis(triethoxysilyl)benzene as described in Refs. [43,44]. The architecture of the hyperconnected network has been achieved through the use of 1,3,5-silyl benzene precursors, where each silicon atom can be connected to the five other nearest silicon neighbors. Thus, the 1,3,5-benzene bridging group structure connects each silicon atom to two others via carbon bridges that share one common Si–C bond while maintaining the ability of a silicon atom to connect with three others via Si–O–Si bonds (Figure 1b in Ref. [33]). The films with different ratio of 1,3,5- and 1,3-benzene bridges were deposited and analyzed.

iv. The films containing only terminal methyl groups (3-1 and 3-2 in Figure 1) were deposited using pure MTMS precursor. Film 3-1 was deposited as “dense” (without sacrificial porogen) and film 3-2 by the addition of 30 wt% Brij^®^30 surfactant C_12_H_25_(OCH_2_CH_2_)_4_OH. The samples deposited with porogen had an open porosity of about 33% as measured by ellipsometric porosimetry (EP) [45], while the “dense” sample had micropores with an open free volume of about 8% [42].

The physical properties of all deposited materials used are summarized in Appendix A. All samples were thermally cured; UV light and plasma were not used for curing.

### 2.2. Analysis

The chemical compositions of the films (1-1, 1-2, 2-1) were analyzed by Nicolet 6700 (Thermo Electron Corporation, Waltham, MA, USA) Fourier transform infrared spectroscopy (FTIR) in the range 4000–400 cm^−1^ with a resolution of 4 cm^−1^ (64 scans) in transmission mode. The optical characteristics of the films, including thickness and refraction index (RI) were measured with a spectroscopic ellipsometer SE850 (Sentech, Berlin, Germany) (λ = 300–800 nm) using the Cauchy polynomial function. The porosity and pore size distribution of the films were characterized by atmospheric pressure EP [45]. Isopropyl alcohol (IPA) vapors (and heptane vapors, in some cases) diluted with dry N_2_ carrier gas in a specially designed bubbler were used as an adsorptive. The open porosity of the films is calculated as the volume of adsorbed liquid adsorbate from RI values measured during IPA adsorption using a modified Lorentz–Lorenz equation [45]:(3)$$\frac{n_{\mathrm{eff}}^{2}-1}{n_{\mathrm{eff}}^{2}+2}=V\frac{n_{\mathrm{ads}}^{2}-1}{n_{\mathrm{ads}}^{2}+2}+\left(1-V\right)\frac{n_{s}^{2}-1}{n_{s}^{2}+2}$$
where n_eff_—is the measured RI of the porous film when the pores are gradually filled by adsorbate at different relative pressures, n_ads_—is the RI of the liquid adsorbate, n_s_—is the matrix RI, and V—is the volume of the condensed adsorbate. The calculation of the pore size distribution (PSD) is based on the analysis of adsorption isotherms. The analysis is based on the Kelvin equation, which describes the dependence of relative pressure (P/P_0_) on meniscus curvature similar to the standard Barrett–Joyner–Halenda (BJH) procedure used in nitrogen adsorption porosimetry [46]. The micropore size uses the Dubinin–Radushkevich approach based on Polanyi’s potential theory of adsorption. EP also allows us to calculate the so-called cumulative surface area. The specific surface area of each small group of pores dA_i_ are calculated from the corresponding pore volume and pore radius as dA_i_ = dV_i_/r_i_ (for cylindrical pores). A value of the cumulative surface area is obtained by assuming the values of dA_i_ over the whole pore system.

UV-induced luminescence of samples with ethylene bridge (1-1 and 1-2) and 1,3,5- and 1,3-benzene bridge (2-1) samples were measured on a FP-8300 spectrofluorometer (JASCO, Tokyo, Japan) using a continuous output Xe arc lamp with shielded lamp housing (150 W) and holographic concave grating in a modified Rowland mount monochromator. The radio-photometer system using monochromatic light was used to monitor the output intensity of the Xe lamp. The samples were mounted in a standard 10 mm rectangular cell holder SCE-846/D061161450 provided by JASCO. The wavelength accuracy and maximum resolution are 1 nm. The excitation and emission spectra are in the range in the energy range from 6.2 to 1.65 eV and the slit width is 5 nm–5 nm. Measurements were performed at room temperature and fully controlled using a Spectra Manager. The excitation and emission bandwidth were 5 nm at a scan speed of 1000 nm/min.

The methyl-terminated (3-1, 3-2) and 1,4-benzene-bridged samples (2-2 and 2-2) were measured at excitation energies of 6 to 10 eV using different systems. Luminescence and luminescence excitation spectra were measured with the photoluminescence end station of the FinEstBeAMS beamline at the 1.5 GeV storage ring of the MAX IV synchrotron facility [47]. The luminescence excitation spectra were measured with a spectral resolution of no less than 4 meV using fused silica and MgF_2_ optical filters in the energy range of 4.5–7.0 eV and 6.5–11 eV, respectively. The samples were placed in an ARS closed-cycle helium cryostat equipped with a LakeShore 325 temperature controller, and the measurement temperature was equal to 7K. Before measurements, the samples were degassed at 350K in a vacuum of 10^−9^ mbar. An excitation flux curve obtained using a factory-calibrated AXUV-100G diode (OptoDiode Corp, Camarillo, CA, USA) was used to correct the excitation spectra. The luminescence spectra were recorded using a fiber-coupled Andor Shamrock SR-303i (Andor Technology Ltd., Belfast, UK) spectrometer equipped with a Hamamatsu H8259-01 photon counting head. The luminescence spectra were corrected for the spectral sensitivity of the registration channel. It is necessary to mention that by using photons with higher energy, we confirmed that we are not missing transitions that cannot be excited by photons with an energy of 6.2 eV. Cryogenic temperature reduces thermal noise and, therefore, should allow to detection of PL peaks of low intensity if they are present. The ability to generate detailed PLE spectra helps to better understand the origin of PL transitions. Using this system, we found a correspondence between the results obtained in these systems, which makes our conclusions more reliable.

It is necessary to mention that all other characteristics of the films, like dielectric constant, mechanical properties, surface roughness, pore ordering, plasma and VUV properties, have also been studied by using different instrumentations. These data are not discussed in this work but can be found in the references [36,37,38,39,40,48,49].

### 2.3. Calculation of the Energy Diagram of UV-Induced Processes

Density Functional Theory (DFT) calculations were carried out systematically, employing the PBE0 density functional [50,51,52]. The way in which the PBE0 functional is derived and the lack of empirical parameters fitted to specific properties make the PBE0 model a widely applicable method for quantum chemical calculations. The molecular geometries, energies and electronic structure of the molecules were studied at the PBE0/6-31G** level of theory, as implemented in the Jaguar 9.6 program [53]. This computational model has already been used successfully for calculations in our previous work [54,55]. Frequency calculations were performed at the same level of theory to obtain the total enthalpy (H) and Gibbs free energy (G) at a standard temperature of 298.15K using unscaled frequencies. The reaction enthalpies (ΔH) and Gibbs free energies of the reaction (ΔG) of the studied molecules were calculated as the difference in calculated H and G between the reactants and products, respectively. The energy of the excited states and oscillator strength (f) of transitions were calculated in the gas phase at the optimized ground state structure of model molecules using the time-dependent (TD) DFT method [56] at the PBE0/6-31G** level of theory. The excited states calculations were performed using the Full Linear Response (FLR) approximation [57], as implemented in the Jaguar 9.6 program. The number of excited states was set to 100 for two reasons: the first is that the initial guess might not accurately reflect the final states; the second is to ensure that the near-degeneracies are accounted for.

## 3. Results

### 3.1. Ethylene-Bridged PMO [39,40]

The FTIR spectra of fully cured ethylene-bridged PMO films (1-1, 1-2) are shown in Figure 2. The absorption band at 1250–1000 cm^−1^ corresponds to Si–O stretching modes. The soft baked (SB) at 150 °C films show the presence of hydroxyl and silanol groups (O–H, Si–OH) at 3700–3100 cm^−1^ and C=O group at 1750 cm^−1^, which are removed after the hard bake (HB). The Si–OH concentration in SB films is higher in the films deposited without or with a small concentration of porogens. The possible reason is that the remaining porogen makes the films more hydrophobic. It can be seen that SB films still contain a significant amount of C–H_x_ (x = 2 or 3) groups in the wavenumber range (3000–2800 cm^−1^) and C–H (1460 cm^−1^). They mainly originate from the template (Brij^®^30), and this is the reason why their concentration increases with initial porogen concentration (Figure 2b). HB reduces those group concentrations (Figure 2d). The presence of Si–CH_3_ terminal groups can be seen from the peak at ~1275 cm^−1^, and Si–CH_3_ groups also contribute to the intensity of the hydrocarbon peaks at ~2970 and ~2920 cm^−1^.

Figure 3 shows the adsorption–desorption isotherms of heptane vapors measured by EP. The isotherms of all samples deposited with 0–30 wt% template do not have a hysteresis loop, indicating that the pores are cylindrical in shape. However, the samples deposited with a 50 wt% template have pronounced hysteresis loops typical for the formation of internal voids leading to “ink bottle”-like effects in isotherms [58].

A clear difference can be seen between the samples deposited with a BMTSE/MTMS ratios of 47/53 and 25/75. The isotherms in samples with a BTMSE/MTMS ratio of 25/75 have critical slopes at higher relative pressures P/P_0_, indicating a larger pore size. They also have higher porosity than the samples deposited with a BTMSE/MTMS ratio of 47/53, and this difference becomes even more pronounced for the samples deposited with 30 and 50 wt% templates (Figure 3). A reasonable explanation is that the ethylene bridge increases the stiffness of the matrix and hinders the agglomeration of template molecules during matrix formation. However, no principal differences in the photoluminescence spectra are observed between 47/53 and 25/75 samples (Figure 4).

The measured luminescence spectra clearly show three peaks located near 4.3, 3.3, and 2.9 eV. The position of these peaks is the same as in the paper [21], where the films of the same series (BTMSE/MTMS = 25/75 ratio, porosity 30%) were studied upon excitation by synchrotron radiation. The only differences relate to the relative intensities of the peaks, but it is likely a sample storage issue. The peaks interpretation in Ref. [21] is based on information reported for SiO_2_. According to [22,23], the peaks at 4.3 and 3.2 eV are often observed in SiO_2_ and can correspond to ODC(I) defects (≡Si–Si≡), and the peak at 2.9 eV was interpreted as emission from ODC(II) defect (=Si:).

In our case, the positions of the 2.9 and 3.2 eV peaks are constant in all three types (AD, SB, HB) of films but the position of 4.3 eV changes during curing. This peak corresponds to 4.3 eV in as-deposited (AD) films and then decreases to ~4 eV in SB samples at 150 °C and HB samples at 400 °C. This shift is too significant to be assigned to only ODC(I). The shift of the 4.3 eV peak, in principle, can be interpreted as the result of the formation of additional peaks at *E* < 4 eV, which can be assigned to specific carbon-containing residues [59,60,61,62,63]. In the general case, the formation of (≡Si–Si≡) type defects should occur due to the destruction of certain bonds of Si atoms. It can be assumed that they can be formed due to the destruction of carbon bridges, which have the lowest thermal stability in this system. However, the –Si–(CH_2_)_2_–Si– bonds are sufficiently stable and certainly cannot be destructed in AD and SB films (*T* < 200 °C). Therefore, the assignment of this peak to ODC(I) (Ref. [21]) is doubtable. The peak at 2.9 eV in BTMSE/MTMS = 47/53 sample increases drastically during SB and remains quite high after HB. It also generates doubt on the assignment to ODC(II) since, according to Ref. [22], ODC(I) is typically much more abundant than ODC(II) because ODC(II) might be a product of a partial transformation of ODC(I).

#### Effect of Storage and ICP Oxygen Plasma

The evolution of PL intensity in a sample with a BTMSE/MTMS ratio of 45/53 (HB) is shown after storage in air in a clean room environment (Figure 5a). The PL intensity gradually increased every week and finally increased ~10 times after 1 month. Figure 5b shows that the first PL measurement of a sample stored after one month reduces the PL intensity by about 10%. The following measurements, one after another, with a break of a few minutes, slightly reduce the intensity.

The corresponding change in chemical composition during sample storage can be seen in FTIR spectra (Figure 6a). Air storage accumulates hydrocarbon residues from the clean room environment. Such an effect is well known for porous low-k materials: capillary forces enhance the adsorption and condensation of hydrocarbons and other residues. The observed decrease in PL intensity in air-stored samples after the first measurement may be related to the UV-induced desorption of part of the carbon residue.

When this sample is exposed to soft oxygen inductively coupled (ICP) plasma (Figure 6b), the concentration of the hydrocarbon-containing species is reduced. At the same time, also some loss of Si–CH_3_ groups and hydrophilization is also observed (see Appendix A Appendix A). The thickness of the low-k film before exposure to O_2_ ICP plasma was equal to 466 nm; then, it reduced to 435 nm after 5 min and 399 nm after 10 min. Meanwhile, the corresponding reductions of Si–CH_3_/Si–O–Si peaks ratio were from 0.173 to 0.163 and to 0.154, respectively. The changes of hydrocarbon residues peaks in the region 2800–3000 cm^−1^ were about three times larger. These observations suggest that the used plasma conditions were sufficiently soft to minimize plasma damage of the low-k matrix (a small change in Si–CH_3_/Si–O–Si ratio), and the ICP-generated oxygen radicals are mainly consumed on the accumulated carbon residue.

Reducing the carbon concentration leads to a reduction in PL intensity at 4.2 eV (Figure 7). The change happening in the PL intensity during storage and exposure to O_2_ ICP plasma clearly shows that the intensity of the peak near 4.1–4.2 eV is related to hydrocarbon residues accumulated during the template removal (pristine sample) and storage in air. PL of hydrogenated carbon has been extensively studied in the past. According to these publications, in addition to the well-known broad PL bands at 2.1–2.33, 2.85–2.92 eV, and 3.17–3.22, other peaks located at 3.64–3.70, 3.93–4.01 and 4.34–4.56 eV in the UV region can also be found [64]. The observed change in PL spectra in the carbon-rich films deposited with high template concentration (50 wt%) and high BTMSE/MTMS ratio (Figure 4) and the increase in the intensity of certain peaks during thermal curing and exposure in O_2_ ICP plasma most probably reflects the modification of carbon-containing components of the studied films.

### 3.2. Benzene-Bridged Organosilica Films

#### 3.2.1. 1,3,5- and 1,3-Benzene-bridged Films [33,43,44]

The films with different molar ratios of 1,3,5-tris(triethoxysilyl)benzene to 1,3-bis(triethoxysilyl)benzene bridging organic groups (1:3 and 1:7) were deposited by spin-on coating followed by a soft bake in the air at 150 °C (SB) and hard bake in the air at 400 °C (HB). The concentrations of the non-ionic template (Brij^®^30) varied from 0 to 41 wt%. The chemical composition of the matrix of the films was evaluated and discussed, as well as refractive indices, mechanical properties, k-values, porosity and pore structure [43,44]. The films containing benzene bridging groups keep the pore size constant and equal to 0.81 nm while their porosity changes from 0 to 30%. The films containing a benzene bridge have a higher Young’s modulus than plasma-enhanced chemical vapor deposition (PECVD) methyl-terminated low-k films with the same porosity [33]. The fabricated films show good stability after a long time of storage. FTIR spectra and porosity data generated by ellipsometric porosimetry are presented in the Appendix A.

Figure 8 shows photoluminescence spectra for 1:3 and 1:7 samples. This means that the 1:3 sample has a 2.3 times higher concentration of 1,3,5-benzene rings than the 1:7 sample. The most important difference from ethylene-bridged films is that AD and SB films contain only 1 pronounced peak at around 3.9 eV. Looking at the PL spectra of samples AD and SB, one can conclude that the intensity increases with increasing porosity. The PL spectra of HB films are becoming more complicated, and new peaks near 2.3, 2.9 and 4.85 eV are observed.

#### 3.2.2. 1,4-Benzene-bridged Films [42]

Two different types of 1,4-benzene-bridged films were prepared. The first one (1,4-BB) was prepared from pure BTESB (1,4-bis(triethoxysilyl)-benzene) without porogen and without methyl terminal precursors and had an intrinsic porosity of about 10%. The second type of film (1,4-BB-p) was also prepared with BTESB, but 30 wt% porogen was added into the precursor. As a result, the film has EP measured porosity of 30%. FTIR and EP data can be found in the Appendix A. All of these samples were fully cured: they went through the SB and HB processes. PL spectra of 1,4-BB film were measured at the photoluminescence station of the FinEstBeAMS beamline at the 1.5 GeV storage ring of the MAX IV synchrotron facility [47]. The measurements were done at 7K after outgassing in an ultrahigh vacuum (10^−9^ mbar) at 350K. PL was excited by UV photons with energy from 4 to 10 nm (PLE). PL spectra of non-porous films have clearly defined bands at 3.67 eV (at excitation by 5.6 eV) and 3.69 eV (at excitation by 7.3 eV) (Figure 9a). The UV damage threshold in OSG low-k films close to 6.0–6.2 eV [19,65].

The results presented in Figure 9a show that PL intensity decreases approximately five times after exposure to 7.3 eV in comparison with the 5.6 eV exposed sample. However, the band position remains the same. A possible explanation is that 7.3 eV photons remove (destruct) part of the component responsible for PL emission at 3.67 eV. Similar conclusions can be made about the porous benzene-bridged films (Figure 9b). These films were irradiated by 5.6, 6.2 and 7.2 eV photons, but the difference in PL bands position is not remarkable. However, the emission intensity drops very much when comparing PL intensity after 6.2 and 7.2 eV excitation in comparison with 5.6 eV. Deconvolution shows the presence of a 2.86 eV peak, and there are also traceable emissions with wavelengths 2.24 and 4.2 eV. The introduction of porogen changes the main emission to 3.76–3.78 eV, and the presence of 4.20 eV is becoming more pronounced (Figure 9b). It supports our previous conclusion that emission at 3.9–4.2 eV is related to hydrocarbon residue: removal of porogen always tends to leave a certain amount of porogen residues [8,9,10]. The similarity of the PL spectra of these samples to the PL spectra of HB samples with 1,3,5/1,3-benzene bridges (Figure 8) shows that the key role is played by the presence of the benzene bridge rather than the type of their bond with Si atoms. Interesting information can also be obtained from PLE spectra (Figure 9c). The most important excitation bands in all cases correspond to 5.8–6.0 eV photons, but the 2.78-2.48 eV emission peaks also have excitation bands at 7.3 eV.

### 3.3. Methyl-Terminated Organosilica Films [66]

The films with only methyl terminal groups were fabricated using pure MTMS to prepare dense films and also with 30 wt% added porogen (MTMS-p). All of these samples were fully cured: they went through the SB and HB processes. FTIR spectra are shown in the Appendix A. The measured porosities were equal to 7.5% and 33.1% (Appendix A). The PL measurements were also done on the system used for 1,4-BB films. MTMS films at 7.0 eV photoluminescence excitation (PLE) showed only one peak with emission at 2.83 eV (Figure 10). The samples excited by UV light with an energy of 6.2 eV show an additional peak at 4.35 eV. The last peak is much more pronounced in the films prepared with a porogen, which also supports our previous conclusion that this peak can be related to the presence of porogen residue. MTMS-p also shows the presence of emission bands at 4.95 and 5.16 eV.

Dense MTMS samples excited with 7.0 eV photons have much lower PL intensity than the sample excited with 6.2 eV photons. The intensity of the 2,83 PL band has significantly reduced, and the band with an energy of 4.35 eV almost disappeared. This also suggests that the light of 7.0 eV is removing the PL component. It should be noted that 6.2 eV UV photons strongly reduce PL intensity in 1.4 BB films but not in the MTMS film. The most likely explanation is the higher UV resistance of the methyl terminal groups in comparison with the carbon bridges of PMO materials [36,37].

The excitation spectra (PLE) are shown in Figure 10b. Photoluminescence at 3.10–2.48 eV in both dense and porous samples have very pronounced excitation bands at 6.2 eV and also shoulders corresponding to 5.5 and 7.25 eV. The 4.59–4.13 emission bands in the dense MTMS film have a selective excitation band at 6.8 eV. The MTMS sample excited by 7.0 eV UV photons shows no emission band at 4.35 eV, and, therefore, it is reasonable to assume that this component is destroyed or removed during the exposure.

## 4. Discussion

The presented results allow us to discuss the origin and nature of the observed luminescence bands. The main results are summarized in Table 1. A clear correlation of PL bands with types of organic groups is observed. 1,4-benzene-bridged films (2-2) show the presence of 3.68 and 3.78 eV PL peaks. Similar emissions at 3.9 eV have demonstrated the films containing both 1,3,5- and 1,3-benzene bridges. It suggests that these emission bands are related to benzene bridges. However, HB (400 °C) films show the appearance of small intensity PL peaks near 2.9 and 4.2 eV. It is reasonable to assume that the appearance of these peaks is associated with the compaction of the film matrix, which occurs during calcination at 400 °C and initiates the segregation of unassembled porogen fragments and other residues from the low-k matrix to the surface of the pore walls [8,41].

The samples with only terminal methyl groups show a PL band at 2.8–2.9 eV. Dense MTMS films only show the presence of this peak when PL is excited by 7.0 eV photons but also have a small peak at 4.2 eV when PL is excited by 6.2 eV photons. The porous MTMS-p films deposited with porogen show an increase in relative intensity of 4.25 eV peak compared to dense materials. The peaks with even higher energy are observed in this sample. An important observation (Table 1) is that all samples deposited with porogen have a strong peak at 4.2–4.3 eV, and this support our conclusion (Section 2.1) that it is related to porogen residues and other hydrocarbon residues.

Ethylene-bridged samples give emissions at 2.9, 3.3 and 4.2 eV. We already showed that PL peaks at 2.9 and 4.2 eV correspond to methyl terminal groups that are present in these films and porogen residues because the sample was deposited with a porogen. Therefore, the 3.3 eV peak reflects the presence of an ethylene bridge.

### Diagrams of the Energy Levels Calculated by Using Density Functional Theory

Figure 11 shows energy level (Jablonski) diagrams calculated for model molecules reflecting the studied materials.

Figure 11a,b show the energy levels for 1,4-benzene and 1,3,5-benzene bridges. After excitation to the first excited singlet S^1^ state, the molecule undergoes ISC to the excited triplet T^1*^ state. The T^1*^ state undergoes further structural relaxation to a more stable T^1^ state due to the adjustment of molecular geometry to change the electron distribution upon excitation. Due to the large energy gap between ground S^0^ and excited S^1^ states, the deactivation of energy from the S^1^ state may be observed by fluorescence. Otherwise, when ISC occurs, deactivation of energy by the phosphorescence of both T^1*^ and T^1^ can be expected. Interestingly, the energy levels of all transitions in the 1,4-benzene and 1,3,5-benzene bridges are very similar despite the difference in bond structure. UV-induced excitation that initiates the singlet–singlet (S^0^–S^1^) transition has energies of 5.35 and 5.22 eV, which is very close to the maximum PLE energy of 5.4–5.5 eV measured for 1,4-benzene-bridged OSG materials. (Figure 9c). The energy of the PL photons measured for these films was equal to 3.7 ± 0.1 eV and 3.9 eV for 1,4- and 1,3,5-benzene-bridged OSG, respectively. The calculated energies for the triplet—singlet (T^1*^–S^0^) transition have very close values equal to 3.94 and 3.91 eV. Taking into account that model molecules were used for calculations, one can conclude that the agreement between the calculated and measured PL and PLE energies is perfect. Therefore, the energy diagrams presented in Figure 11a,b most likely correctly reflect the electronic transitions leading to PL.

Figure 11c shows the energy diagram for a molecule having both ethylene bridge and methyl terminal groups. After excitation to the first excited singlet S^1^ state, molecules undergo ISC into the excited triplet T^1*^ state. However, the T^1*^ state is very unstable due to the high energy and undergoes Si–O bond cleavage during optimization instead of relaxing into a more stable T^1^ state. Due to the large energy gap between ground S^0^ and excited S^1^ states, deactivation of the energy from the S^1^ state through fluorescence may be observed. Otherwise, if ISC into T^1*^ state occurs, deactivation of the energy from T^1*^ through phosphorescence may be expected. The calculated energy for the singlet–singlet (S^0^–S^1^) transition for this molecule (6.26 eV) is quite close to the measured PLE energy for the sample 1-2 (BTMSE/MTMS 47/53), equal to 6.2 eV (Appendix A). However, we do not observe the emission of 6.19 eV corresponding to transition T^1*^–S^0^. Moreover, after the relaxation of T^1*^ to the T^1^ state, this molecule dissociates, forming the radicals shown in Figure 11c. Nevertheless, the bond dissociation energy (BDE) is equal to 3.6 eV, which is close to PL observed with this compound ~3.3 eV. A simulation by Density Functional theory makes this radiative transition doubtful because of energetically preferential molecular destruction from the triplet state. It can be assumed that the destruction of these molecules occurs in parallel with UV radiation, and therefore the intensity of the characteristic 3.3 eV emission in the films with an ethylene bridge has a relatively low intensity (Figure 4).

Figure 11d,e show the energy diagrams for two different molecular structures representing methyl-terminated OSG materials. The first one reflects a model SiO_2_-like structure where one Si bond is terminated by the methyl radical. In the second structure, we selected tetramethylcyclotetrasiloxane as a model to take into account the possibility of the second (hydrogen) terminal group being bonded to silicon. However, the energy diagrams in these two cases were very similar: 7.07 eV for the S^0^–S^1^ transition and 6.83 eV for the T^1*^ state. Only a small difference can be seen in BDE energy: 4.24 eV and 4.13 eV, respectively. It is obvious that the calculated values are quite different than the measured ones (6.2 eV for PLE and 2.8 eV for PL), although the PLE spectrum contains a band at 7.25 eV able to provide a S^0^–S^1^ transition in these molecules. The most important feature is that according to the DFT calculations, molecules in the T^1*^ state become unstable and dissociate to form radicals, as shown in Figure 11d,e. The most reasonable assumption that can explain the observed PL bands 2.78 eV in samples 3-1 and 3-2, and also the PL bands of 2.9 eV in samples 1-1 and 1.2 is PL of dissociation by-products. This mechanism needs further investigation. However, PL measurements of the MTMS samples were carried out at 7K, while the DFT calculations correspond to room temperature. Our estimations showed that temperature has little impact on the BDE, while the change in the Gibbs free energy of the reaction is significant. The dissociation of the Si–CH_3_ bond in S^0^ ground state DG(298K) = 78 kcal/mol and DG(7K) = 90 kcal/mol. For the dissociation from the triplet state DG(298K) = –104 kcal/mol and DG(7K) = –86 kcal/mol. Therefore, the probability of the Si–CH_3_ dissociation reaction at 7K is much lower and therefore, parallel UV emission can also be expected. Similar phenomena were reported in the papers [67,68]. The calculated energy of the emission is about 4.2 eV, which overlaps with the peak attributed to carbon residue radiation. For this reason, these peaks are observed in all samples deposited with porogen and containing CH_3_ terminal groups.

Therefore, the characteristic PL emissions of mesoporous organosilica films correspond to the introduced carbon fragments. It is most likely that they do not include emissions from oxygen-deficient centers (ODC), as was concluded in a recent publication [21]. This conclusion is based on our sample preparation strategy: the samples were not exposed to high-energy impacts capable of generating oxygen vacancies (no UV curing, no exposure to ion and plasma radiation). The curing temperature was not higher than 430 °C, and it is too low for the formation of oxygen-deficient centers [22]. It should be noted that our conclusion is consistent with the results of the ESR studies of various OSG low-k materials [17,32,69,70,71,72,73], which show the presence of carbon dangling bond-related signals in low-k films deposited using sacrificial porogen. A more detailed discussion of these defects and their effect on electrical properties can be found in the review [19] and the corresponding references cited in this paper under the numbers (306, 307, 316–324). The presence of carbon in the film in various forms ranging from the terminal and bridging carbon groups to clusters of elemental carbon originating from porogen or template residues, can give rise to deep energy levels in the insulator bandgap that causes low-field leakage currents [9,10,74].

Finally, similar conclusions were drawn based on the results of studying the PL in SiC_x_O_y_ films deposited using thermal CVD [70]. Films deposited in this way should have properties similar to our MTMS (3-1, 3-2) films, which contain random porosity and methyl terminal groups. Using a parallel study of PL and ESR, the authors concluded that typical structural defects in the oxides, e.g., Si-related neutral oxygen vacancies or non-bridging oxygen hole centers, cannot be considered the dominant mechanism for the white luminescence from SiC_x_O_y_. It was concluded that PL from SiC_x_O_y_ thin films can result from the generation of carriers due to electronic transitions associated with the C–Si/C–Si–O bonds during optical absorption, followed by recombination of these carriers between energy bands and in their tail states associated with Si–O–C/Si–C bonds. Although the detailed mechanism may differ from ours, the key importance of the Si–C and Si–O–C bonds is also emphasized.

## 5. Conclusions

The UV-induced photoluminescence of organosilica films with various combinations of ethylene and benzene bridging groups in their matrix and terminal methyl groups on the pore wall surface was studied to reveal optically active defects and understand their origin and nature. The careful selection of the films’ precursors and conditions of deposition and curing, excluding breakage of chemical bonds and analysis of chemical and structural properties, led to the conclusion that luminescence sources are not associated with the presence of oxygen-deficient centers, as in the case in pure SiO_2_, and has also been predicted for low-k organosilica films [21,22,23]. It is shown that the sources of luminescence are the carbon-containing components that are part of the low-k matrix (3 different types of benzene bridges), as well as the carbon residues formed upon removal of the template and destruction of organosilica samples containing ethylene bridge and methyl terminal groups. A fairly good correlation between the energy of the photoluminescence peaks and the chemical composition is observed (Table 1). This correlation is also confirmed by the results of density functional theory calculations. The spectra become more complicated after annealing at 400 °C, although Fourier transform infrared spectroscopy does not show these changes. The appearance of new peaks after annealing at 400 °C is associated with the segregation of hydrocarbon residues from the low-k matrix on the pore wall surface [8,41]. Therefore, photoluminescence spectroscopy can potentially be used as effective instrumentation to study the modification of these films. The use of photoluminescence spectroscopy can be important to understand the optical and electrical properties and reliability of integrated low-k dielectrics.

Photoluminescence intensity depends on the internal surface area. The correlation between the photoluminescence intensity of the 4.3 eV peak and the measured surface area calculated for the films measured immediately after full curing can be seen in Appendix A. Further investigations of luminescence sources are planned, including their correlation with dielectric properties, the reliability of low-k dielectrics, and their evolution during various technological treatments, including the use of plasma, UV light and ion irradiation.

## Figures and Tables

**Figure 1 nanomaterials-13-01419-f001:**
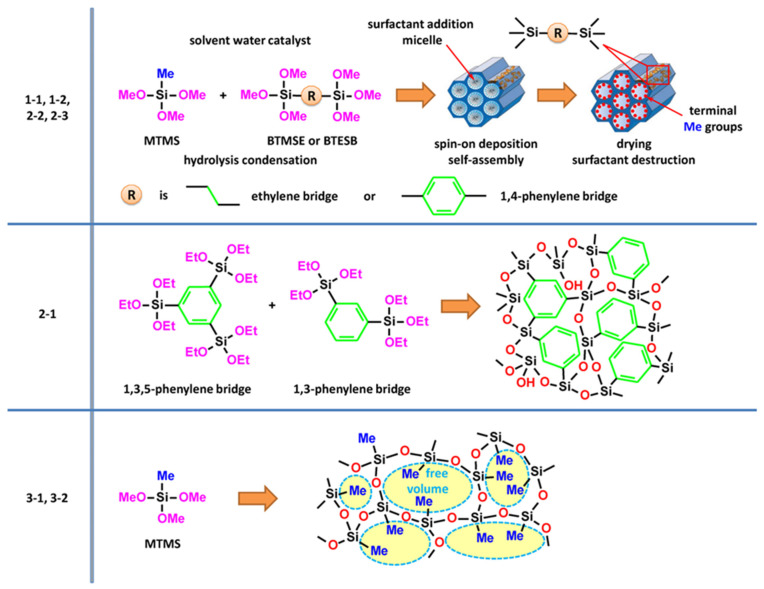
Schematic presentation of the samples used in this research where, Me is a methyl group (–OCH_3_), OMe is a methoxy group, OEt is an ethoxy group (–OCH_2_CH_3_), MTMS is methyltrimethoxysilane, BTMSE is 1,2-bis(trimethoxysilyl)ethane, BTESB is 1,4-bis(triethoxysilyl)benzene. First type samples (1-1, 1-2, 2-2 and 2-3): Ethylene and 1,4-benzene bridged PMO materials with ordered porosity and methyl terminal groups on pore wall surface. Second type sample (2-1): Hybrid OSG films containing both 1,3,7-benzene and 1,3-benzene bridges. Third type samples (3-1, 3-2): methyl terminated OSG materials.

**Figure 2 nanomaterials-13-01419-f002:**
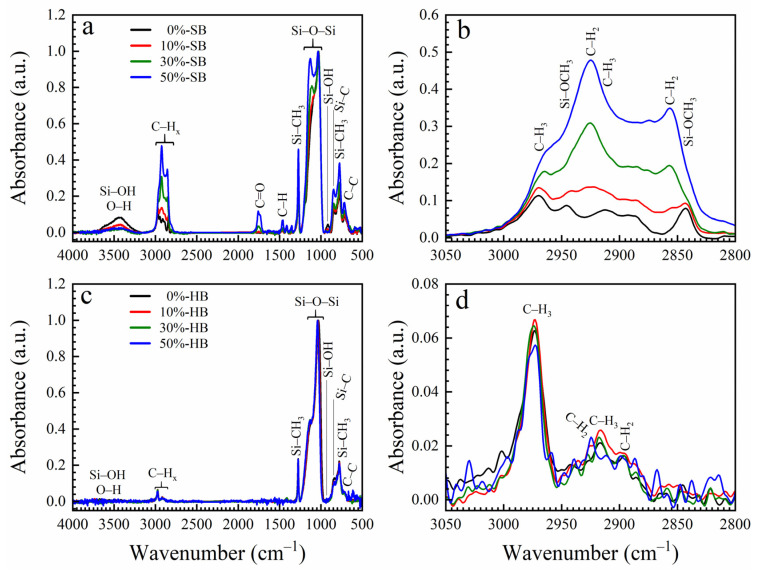
FTIR spectra of OSG low-k samples deposited with a BTMSE/MTMS precursor ratio of 25/75 and different porogen concentrations (0–50 wt%) after soft bake at 150 °C (SB) in the air (**a**,**b**) and hard bake at 400 °C (HB) in the air (**c**,**d**). FTIR spectra of the samples deposited with a BMTSE/MTMS ratio of 47/53 are qualitatively similar.

**Figure 3 nanomaterials-13-01419-f003:**
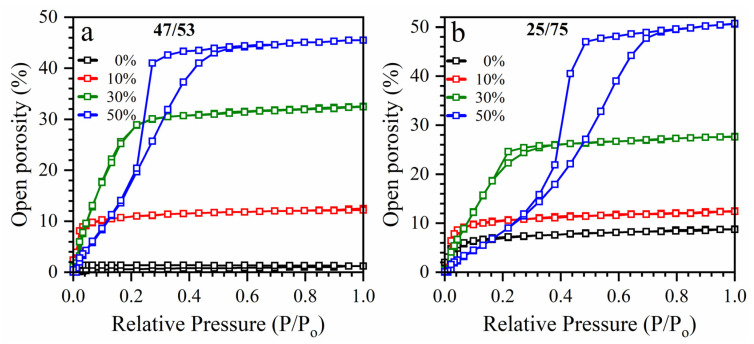
Adsorption–desorption isotherms of heptane vapor in the OSG low-k films deposited with BMTSE/MTMS ratio of 47/53 (**a**) and 25/75 (**b**) and different porogen concentrations (0–50 wt%).

**Figure 4 nanomaterials-13-01419-f004:**
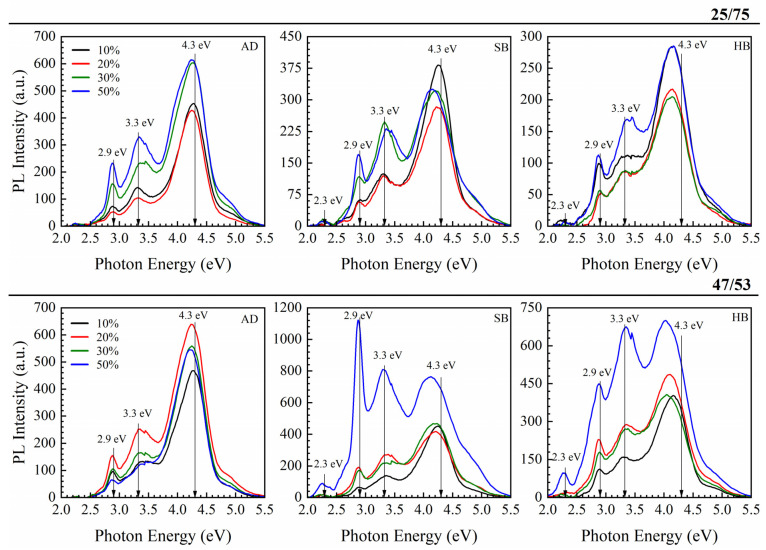
UV-induced room temperature luminescence of PMO low-k films containing both methyl terminal and ethylene bridging groups. The films have different porosity and were cured in air. The samples were deposited with a BTMSE/MTMS ratio of 25/75 (top line) and 47/53 (bottom line) with different porogen concentrations (10–50 wt%) as deposited (AD) after soft bake at 150 °C (SB) in air and hard bake at 400 °C (HB) in air, upon excitation with light of 6.2 eV. The data are reproduced from the paper “Effect of methyl terminal and ethylene bridging groups on porous organosilicate glass films: FTIR, ellipsometric porosimetry, luminescence dataset” by Md Rasadujjaman, J. Zhang, K. P. Mogilnikov, A. S. Vishnevskiy, J. Zhang, M. R. Baklanov (Data in Brief 35 (2021) 106895) with granted permission from ELSEVIER [40].

**Figure 5 nanomaterials-13-01419-f005:**
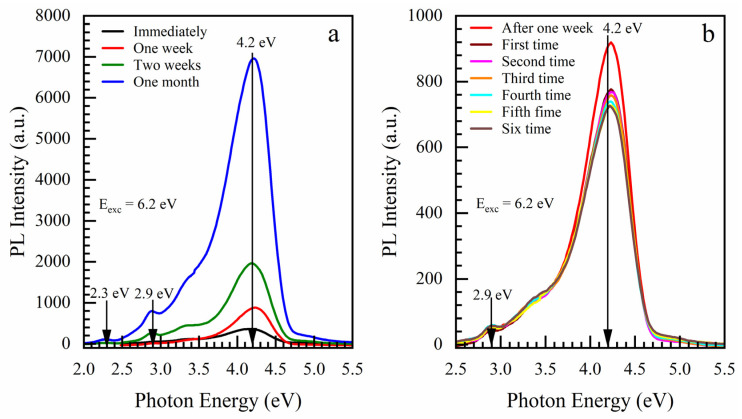
(**a**) Effect of storage in a clean room environment on the PL intensity for the OSG films deposited with a BTMSE/MTMS ratio of 47/53 and (**b**) PL measurements one after the another with a break of a few minutes.

**Figure 6 nanomaterials-13-01419-f006:**
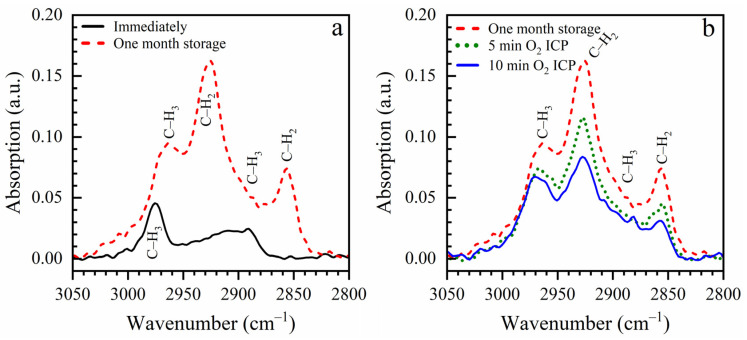
Accumulation of hydrocarbon residues from a clean room environment (**a**) and removal of these residues by ICP oxygen plasma (**b**).

**Figure 7 nanomaterials-13-01419-f007:**
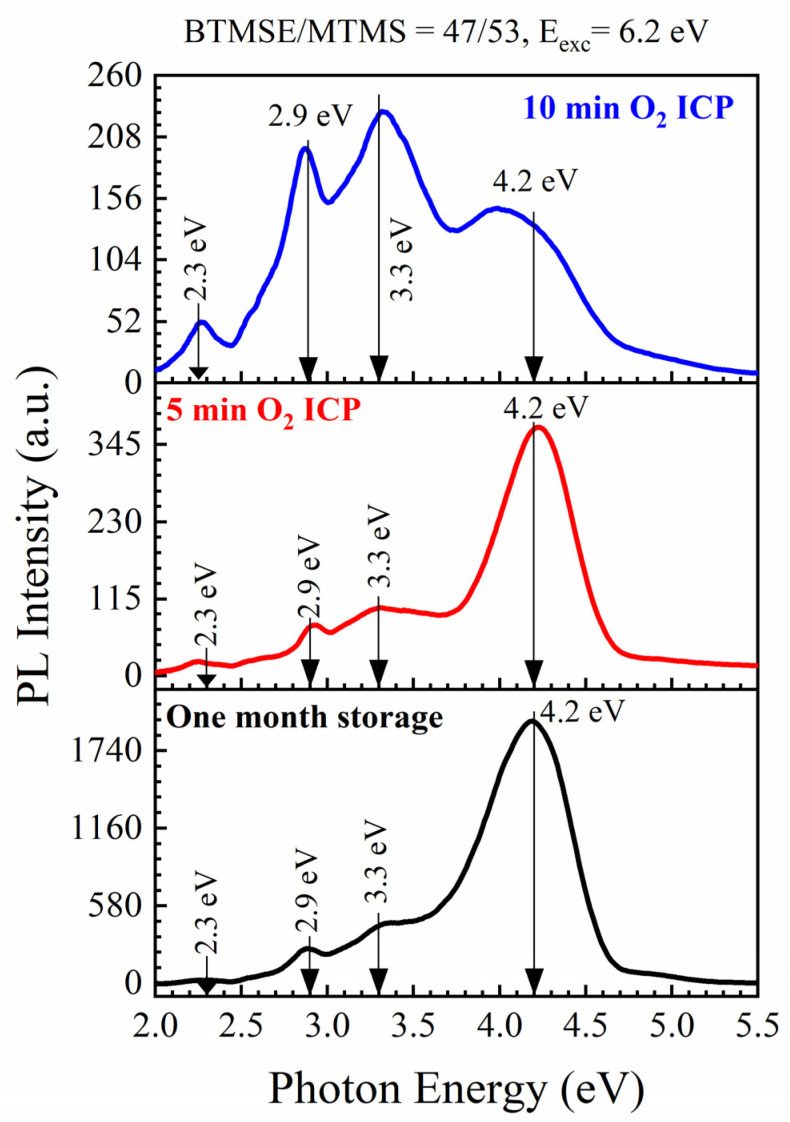
Reduction of 4.2 eV PL intensity during exposure to ICP oxygen plasma. The excitation energy of the photon was 6.2 eV.

**Figure 8 nanomaterials-13-01419-f008:**
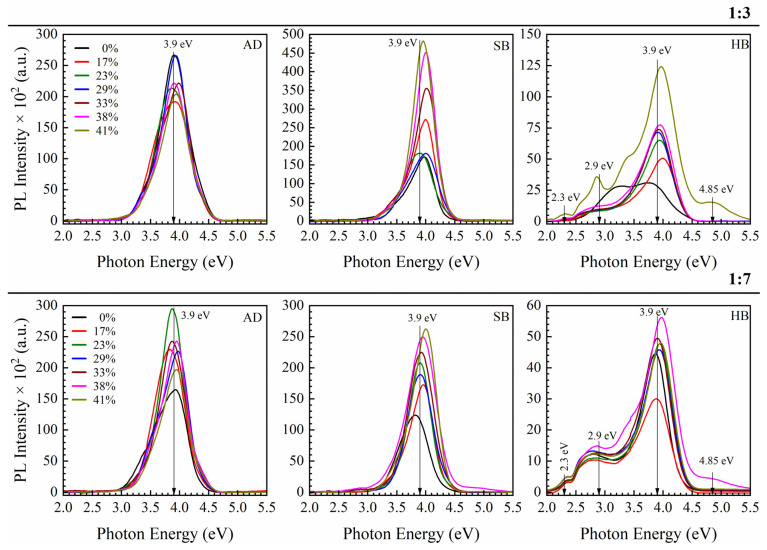
Photoluminescence spectra of OSG films containing 1,3,5- and 1,3-benzene bridges (samples 2-1 in Figure 1) and different porosity upon excitation with light of 6.2 eV. The porogen concentration varied from 0 to 41 wt%. The ratio of 1,3,5- and 1,3-benzene bridges is equal to 1:3 on the top line and 1:7 on the bottom lines. AD, SB and HB have the same meaning as in the samples with ethylene bridge (Figure 4).

**Figure 9 nanomaterials-13-01419-f009:**
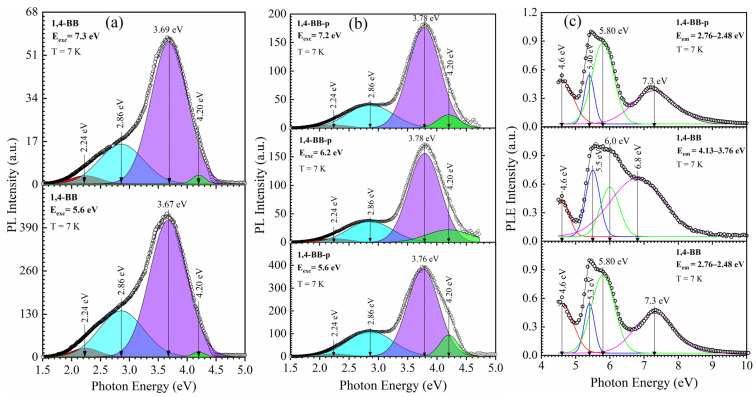
PL spectra of non-porous hard baked OSG film with 1,4-benzene (1,4-BB) bridge upon excitation by 5.6 and 7.3 eV photons (**a**) and porous films with 1,4-benzene bridge (1,4-BB-p) upon excitation by 5.6, 6.2 and 7.2 eV (**b**) and PL excitation spectra (PLE) for 1,4-BB film upon detection 2.76–2.48 eV and 4.13–3.76 eV and for 1,4-BB-p film upon detection 2.76–2.48 eV (**c**).

**Figure 10 nanomaterials-13-01419-f010:**
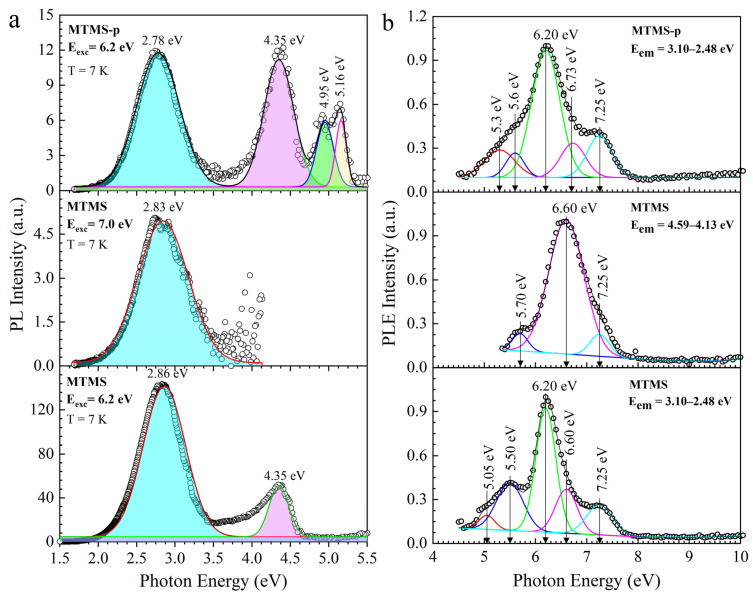
PL spectra of non-porous hard-baked OSG film with methyl terminal group (MTMS) upon excitation by 6.2 and 7.0 eV photons and porous methyl-terminated film (MTMS-p) upon excitation by 6.2 eV (**a**) and PL excitation spectra of MTMS films upon detection 3.10–2.48 and 4.59–4.13 eV and MTMS-p films upon detection 3.10–2.48 eV (**b**).

**Figure 11 nanomaterials-13-01419-f011:**
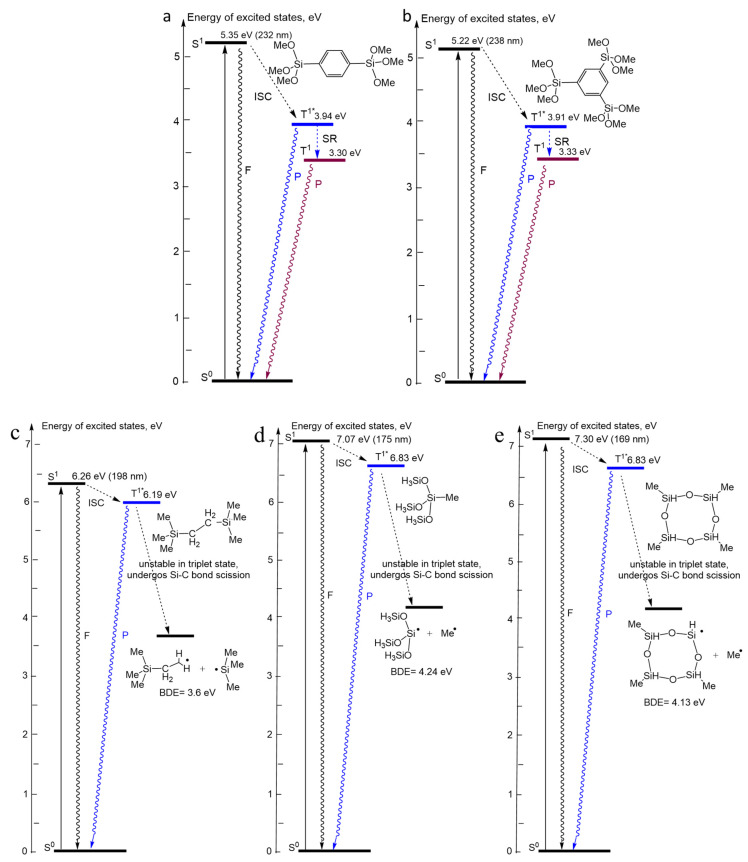
Calculated Jablonski diagram for model molecules containing: 1,4-benzene bridge (**a**), 1,3,5-benzene bridge (**b**), ethylene bridges (**c**), methyl-terminated (**d**,**e**). (F) is fluorescence, (P) is phosphorescence, ISC—intersystem crossing, SR—structural relaxation.

**Table 1 nanomaterials-13-01419-t001:** General characteristics of used OSG films and PL energy as a function of the carbon groups. For example, peaks at 3.78–3.9 eV are only observed in films with benzene bridges, so they can be attributed to the benzene bridge. A similar approach is used for other groups. The brackets show the low-intensity peaks, which appear mainly after HB.

**Sample Number**	**Sample ID**	**Terminal Group** **(–CH_3_)**	**Bridging Group**	**Porogen**	**Characteristics PL Peaks (eV)**			
1-1	BTMSE/MTMS = 47/53	YES	Ethylene	YES	-	3.3	2.9	4.2
1-2	BTMSE/MTMS = 25/75	YES	Ethylene	YES	-	3.3	2.9	4.2
2-1	1,3,5/1,3-BB	NO	Benzene	YES	3.9	-	(2.90)	(4.85)
2-2	1,4-BB	NO	Benzene	NO	3.68	-	(2.86)	-
2-3	1,4-BB-p	NO	Benzene	YES	3.78	-	(2.86)	4.2
3-1	MTMS	YES	NO	NO	-	-	2.85	(4.35)
3-2	MTMS-p	YES	NO	YES	-	-	2.78	4.35
Assignment of PL peaks					Benzene bridge	Ethylene bridge	–CH_3_ terminal	C–H_x_

## Data Availability

The data presented in this study are available on request from the corresponding author.

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
