# Peer review of "UV-Excited Luminescence in Porous Organosilica Films with Various Organic Components"

_nanomaterials, 2023, doi:10.3390/nano13081419_

Round 1
Reviewer 1 Report
The authors have submitted a very thorough investigation of photoluminescence (PL) in porous organo-silica films prepared by spin-on deposition methods. The authors stated objective is to determine the origin of the observed PL and specifically delineate between whether the various carbon moities present organo-silica films are responsible or whether oxygen deficient centers (silicon dangling bond) defects are the source of PL. The support their hypothesis that the various forms of carbon in organo-silica is in fact responsible for the observed PL, the authors examine an impressive range of organo-silica films with varying organic linkers (ethyl vs. phenyl), terminal methyl concentrations, and porosity (i.e. organic porogen loading). They further vary the relative concentrations of all these entities by examining these spin-on films immediately after coating, soft bake, and hard bake. The end result, is that the authors are able to solidly support their hypothesis that carbon in organo-silica films is responsible for the PL and are able to further correlate the energy of different PL emission to specific carbon entities in the film. This reviewer is highly impressed with the article and looks forward to its eventual publication. This said, there are a few areas where the article needs further improvement and corrections in order to ensure this article achieves its potential impact. Please consider the following:
1. For one set of organo-silica films, the authors use a Xe arc lamp for PL measurements, but for another completely different set the authors use a synchrotron and cryogenic temperatures. No explanation is given for why. The authors need to elaborate on this. Was the synchrotron necessary to access PL levels deeper in the UV or for higher resolution; or was there tool availability issues that necessitated using two different instruments. Either way, the authors need to explain this and discuss any potential differences that may result from using the two different instruments or performing measurements at cryogenic temperatures.
2. Figures 9 and 10 show PL spectra in two different energy ranges. Specifically, one panel shows PL from 1.5 to 5 eV and then another panel shows PL from 4 - 10 eV. The text discusses the PL in the 1.5 to 5 eV range, but no (or very little mention) is made of the PL in the higher energy range. The authors need to elaborate further on this and specifically mention in the main text the additional panels shown in Figures 9 and 10.
3. In the discussion part of the article, the authors attempt to connect their PL measurements to prior electron paramagnetic resonance (EPR) measurements on similar materials. Unfortunately, the review paper they cite here is inappropriate as it has nothing to do with EPR and makes no mention of the 306, 307, 316-321 samples they cite. The authors need to double check that they are citing the intended article.
4. Along these lines, the authors don't cite some key articles on EPR measurements of organo-silica films which may or may not be the articles they intended to cite above. In case the authors have simply missed these articles in their literature search, please consider the following:
- Pomorski, J. Appl. Phys. 115, 234508 (2014).
- Mutch, J. Appl. Phys. 119, 094102 (2016).
- Bittel, Appl. Phys. Lett. 97, 063506 (2010).
5. The above Bittel article illustrates how UV radiation can actually create defects in low-k organo-silica films. The reviewer appreciates that the authors did not use UV curing in the processing of their organo-silica films, but they did shine pretty intense UV light on the samples during the PL measurements. Accordingly, the authors should discuss what impact the UV light during the PL measurement might have had on the films during measurement and whether any defects were created during the measurements.
6. Lastly, this reviewer encourages the authors to attempt to concisely summarize the results of this article into a single figure illustrating the various different PL emissions lines they observed and correlated to specific organic motifs in the organo-silica films. Perhaps something similar to Figure 17 in the Mutch (ref. 20) article that they cite.
Reviewer 2 Report
The manuscript from Rasadujjaman et al. reported on the synthesis of organosilica film with ethylene and benzene bridging groups. Moreover, the UV induced photoluminescence of the organosilica films was studied and the present manuscript aims to study the origin of ultra-violet (UV) induced PL in porous organosilica films. The deposited films had well-defined chemical composition and porosity in order to understand the physical nature of optically and electrically active defects.
In conclusion, this is an interesting manuscript which contributes to understand the optical and electrical properties and reliability of integrated low-k dielectrics. In my opinion, the manuscript deserves to be published. However, few details are required concerning the method of the film deposition. Indeed, the procedure to perform the deposition of the film is not described in the method section. Moreover, the thickness of the film required to be determined and reported as some physical and chemical properties are subject to change with film thickness such as the effect in depth of the exposure to O2 ICP plasma.
Reviewer 3 Report
The manuscript reports a thorough investigation of the origin of the photoluminescence emission of porous organosilica films with various organic components. The manuscript is well written, the data is clearly presented, and conclusions are consistent with the data. The manuscript shows that there is a clear correlation between the nature of the organic components and the observation of characteristic PL bands. However, I do feel that this manuscript would be better suited for the sister journal Materials, as the PL of the films does not exhibits quantum confinement effects nor it shows any size dependence. The PL does not depend on the nanoscale thickness of the film nor on the porosity, but rather on the chemical composition. For the sake of completeness, I would recommend the authors to measure the the emission lifetime to support the assignment of the PL characteristic peaks of the spin forbidden transitions. Information on the procedure for the TD-DFT calculations should also be included in a revised version.
Round 2
Reviewer 3 Report
The quality of the manuscript has been improved upon revision. I recommend accepting the manuscript in its present form.